environmental science, evolution, palaeontology

Euarchontoglires, lagomorphs, Palaeogene, endocast, evolution

**Author for correspondence:**
Łucja Fostowicz-Frelik
e-mail: lfost@twarda.pan.pl

# Cranial endocast of the stem lagomorph *Megalagus* and brain structure of basal Euarchontoglires

Sergi López-Torres[1,2,3,4], Ornella C. Bertrand[5], Madlen M. Lang[6], Mary T. Silcox[6] and Łucja Fostowicz-Frelik[4,7,8]

[1]Division of Paleontology, American Museum of Natural History, New York, NY, USA
[2]Richard Gilder Graduate School, American Museum of Natural History, New York, NY, USA
[3]New York Consortium in Evolutionary Primatology, New York, NY, USA
[4]Department of Evolutionary Paleobiology, Institute of Paleobiology, Polish Academy of Sciences, Warsaw, Poland
[5]School of Geosciences, Grant Institute, University of Edinburgh, Edinburgh EH9 3FE, UK
[6]Department of Anthropology, University of Toronto Scarborough, Toronto, Ontario, Canada
[7]Key Laboratory of Vertebrate Evolution and Human Origins, Institute of Vertebrate Paleontology and Paleoanthropology, Chinese Academy of Sciences, Beijing 100044, People's Republic of China
[8]CAS Center for Excellence in Life and Paleoenvironment, Beijing 100044, People's Republic of China

SL-T, 0000-0002-0046-1013; OCB, 0000-0003-3461-3908; MML, 0000-0003-2604-4733; MTS, 0000-0002-4174-9435; ŁF-F, 0000-0002-1266-1178

Early lagomorphs are central to our understanding of how the brain evolved in Glires (rodents, lagomorphs and their kin) from basal members of Euarchontoglires (Glires + Euarchonta, the latter grouping primates, treeshrews, and colugos). Here, we report the first virtual endocast of the fossil lagomorph *Megalagus turgidus*, from the Orella Member of the Brule Formation, early Oligocene, Nebraska, USA. The specimen represents one of the oldest nearly complete lagomorph skulls known. Primitive aspects of the endocranial morphology in *Megalagus* include large olfactory bulbs, exposure of the midbrain, a small neocortex and a relatively low encephalization quotient. Overall, this suggests a brain morphology closer to that of other basal members of Euarchontoglires (e.g. plesiadapiforms and ischyromyid rodents) than to that of living lagomorphs. However, the well-developed petrosal lobules in *Megalagus*, comparable to the condition in modern lagomorphs, suggest early specialization in that order for the stabilization of eye movements necessary for accurate visual tracking. Our study sheds new light on the reconstructed morphology of the ancestral brain in Euarchontoglires and fills a critical gap in the understanding of palaeoneuroanatomy of this major group of placental mammals.

## 1. Introduction

Euarchontoglires, one of the four modern placental clades recognized both in multigene (e.g. see [1]) and morphological studies (e.g. [2–4]), consists of Euarchonta (primates, scandentians, dermopterans and their fossil relatives) and Glires (lagomorphs, rodents and their kin; figure 1). The fossil record of Euarchonta begins with the North American *Purgatorius*; its first appearance is in the earliest Palaeocene, Puercan 1 North American Land Mammal Age (NALMA) [7]. The earliest record of Glires is Chinese *Mimotona*, a duplicidentate representative of the cohort (i.e. the group that includes all Glires related more closely to *Lepus* than *Mus*) [6], dated at 61.0 Ma, Shanghuanian Asian Land Mammal Age (ALMA), equivalent to Tiffanian 2 NALMA [8], which makes the fossil evidence for Duplicidentata only slightly less ancient than that for Euarchonta.

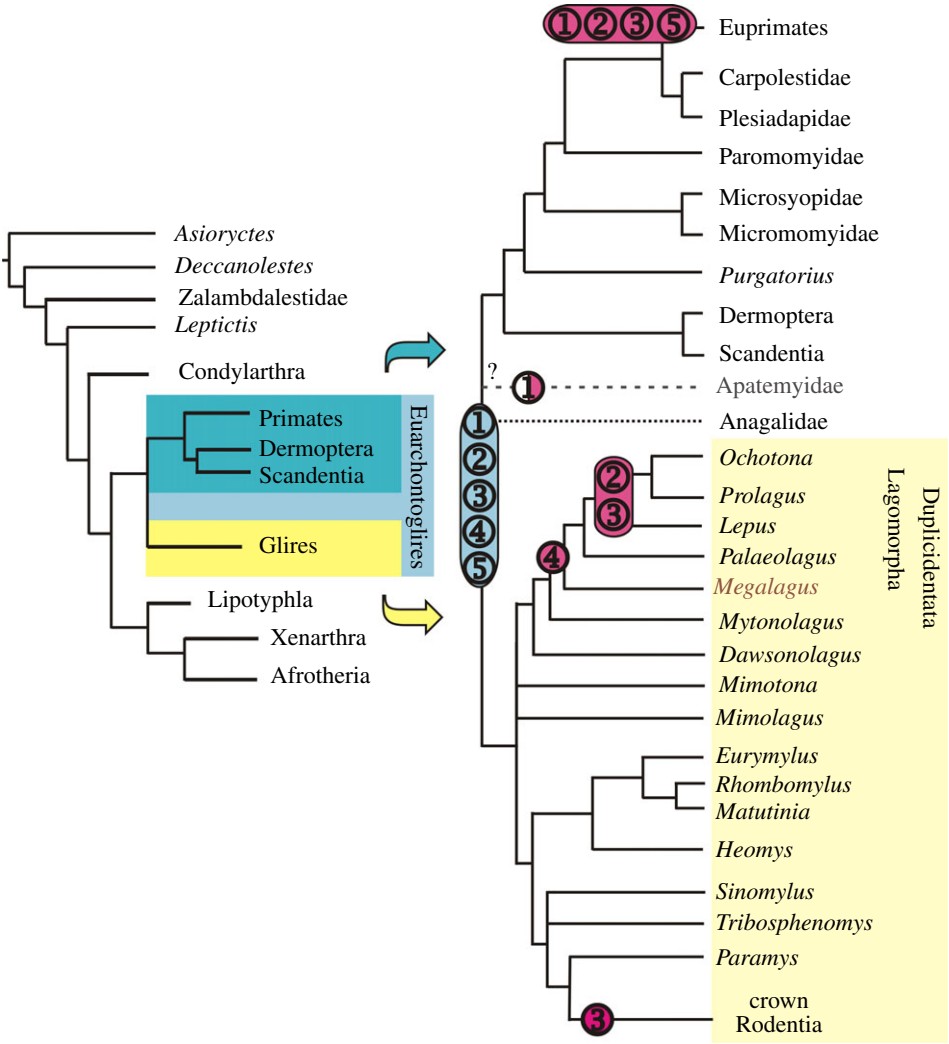

**Figure 1.** Phylogenetic scheme of the Euarchontoglires relationships. Based on Meng *et al*. [5], Silcox *et al*. [4] and Fostowicz-Frelik & Meng [6], modified. Primitive and derived characters (in blue and pink, respectively): (1), size of the olfactory bulbs (blue, large; pink, small; white, extra large); (2), midbrain exposure (blue, large; pink, small); (3), neocorticalization (blue, small; pink, increased); (4), petrosal lobules (blue, small; pink, large); (5), encephalization quotient (EQ; blue, low; pink, high). (Online version in colour.)

However, there is very little fossil evidence for cranial structure in Lagomorpha prior to the late Eocene, with the sole exception of *Dawsonolagus antiquus* from the lower part of the Arshanto Formation (late Early Eocene) of Nei Mongol, China, represented by a partial skull with a completely absent basicranium [9]. Following the first radiation of the group in the early middle Eocene of central Asia (Irdinmanhan ALMA), lagomorphs quickly appeared in North America, where they have been present since the middle Eocene (*ca.* 42 Ma, late Uintan NALMA, [10]). By the end of the Eocene (Chadronian), North American lagomorphs were quite diverse (e.g. [10–13]). One of these lineages is represented by *Megalagus*, from the Eocene and Oligocene of North America [13]. *Megalagus* is a member of an early-branching lineage of stem lagomorphs; its closest relatives, *Tachylagus* and *Mytonolagus,* can be traced back to the middle Eocene (late Uintan NALMA) [10,14]. As such, *Megalagus* is the most basal lagomorph taxon for which the complete skull is known [6,13].

In this paper, we use high-resolution X-ray computed tomography (CT) data to provide, to our knowledge, the first description of a lagomorph digital endocast, based on the well-preserved skull of *Megalagus turgidus*. The evolutionary history of the lagomorph brain is almost entirely unknown, in contrast with our knowledge of other groups within Euarchontoglires (e.g. see review in [15]). Only the brain of the European rabbit (*Oryctolagus cuniculus*) has been studied in detail, using magnetic resonance imaging (MRI, e.g. [16] and [17]). However, this laboratory species represents a fairly recent (*ca.* 5.0 Ma for the genus) crown leporid radiation [18]. Previously published natural endocasts [19,20] of fossil lagomorphs also pertain to quite recent (Pliocene) taxa. As such, the *Me. turgidus* specimen studied here gives us the best available insight into the endocast morphology of early lagomorphs and yields data previously missing for this branch of Glires.

Basal duplicidentates branched off before the lineages of Rodentiaformes (i.e. *Tribosphenomys* and other Alagomyidae; see [21]) and paramyine rodents (figure 1; see also e.g. [22] and [5]: figure 74). Morphologically, duplicidentates (including lagomorphs) are in many ways the most basal and conservative modern members of Glires (e.g. [5,22,23]). For these reasons, the endocast of *Megalagus* may provide crucial information on the condition of the brain near the split between basalmost Glires and Euarchonta. This makes our study of the endocast of *Megalagus* pertinent not only to understanding brain evolution in Glires, but also in Euarchontoglires.

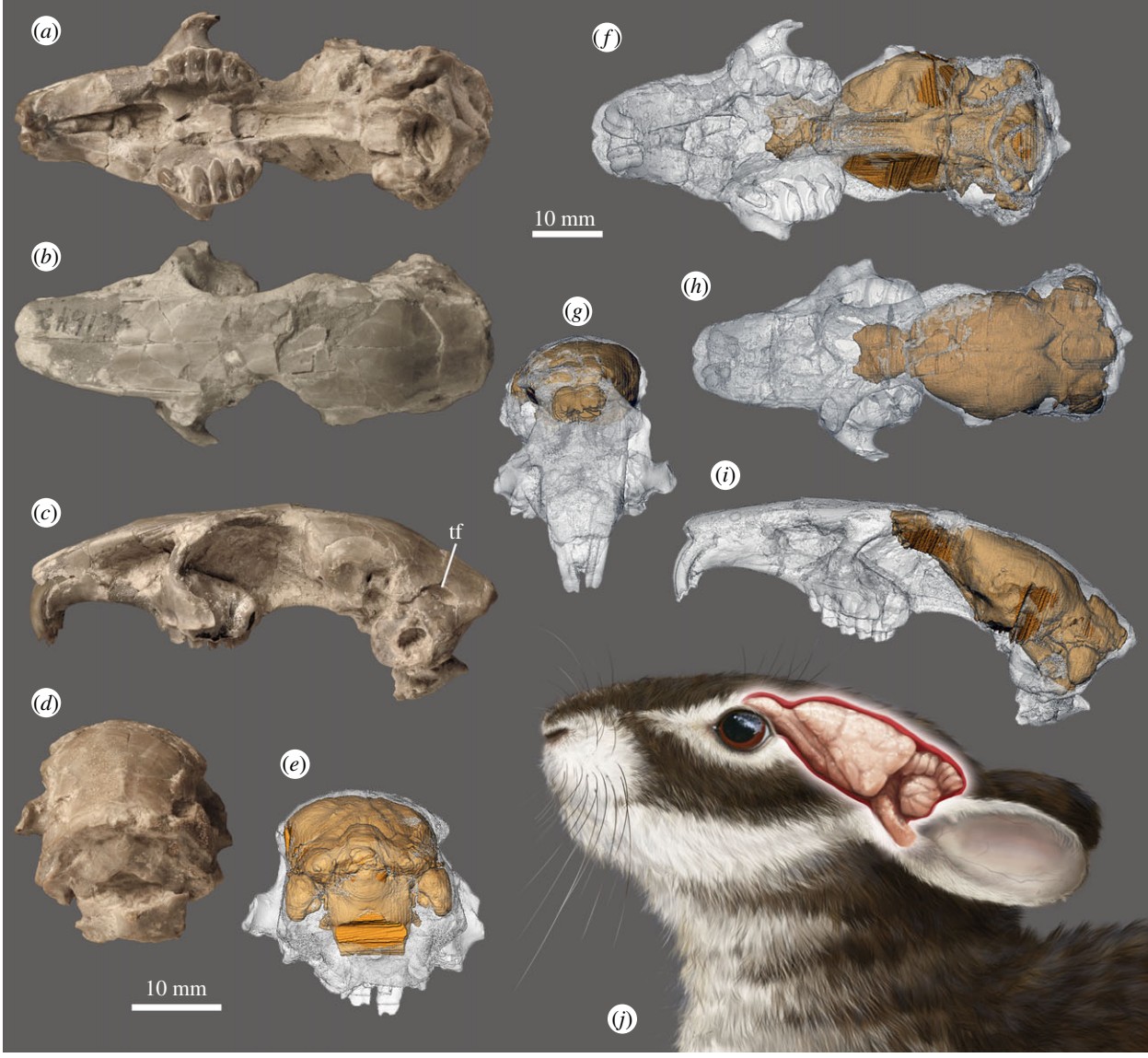

**Figure 2.** Cranium and endocast visualization of *Megalagus turgidus* (FMNH UC 1642) from the Brule Formation at Grime's Ranch, Sioux County, Nebraska. (*a–d*) External cranial morphology, (*e–i*) virtual endocast location in the cranium, and (*j*) life reconstruction of the *Megalagus* head with brain visualized (artist: Agnieszka Kapuścińska). (*a,f*) ventral; (*b,h*) dorsal; (*c,i,j*) lateral; (*d,e*) caudal; and (*g*) frontal views. Abbreviation: tf, temporal foramen. (online version in colour.)

## 2. Description

The endocast of *Megalagus turgidus* was extracted (see the electronic supplementary material) from high-resolution X-ray CT data of an almost complete (only the zygomatic arches are missing), undistorted skull (FMNH UC 1642; figure 2), dated to the earliest Orellan (early Oligocene; 33.7–32.00 Ma) [24] from the Brule Formation at Grime's Ranch, Sioux County, Nebraska [25].

The endocast is elongate, with well-developed, ovoid and pedunculated olfactory bulbs (figure 3), which extend in the cranium to the area above the upper M1 and cover approximately two thirds of the length of the orbit (figure 2). The cerebral hemispheres are oval, gently rounded at the sides, slightly tapering anteriorly, with the maximum width near its mid-length. The frontal lobes in *Megalagus* do not overlap the olfactory bulbs and the circular fissure is relatively wide (figure 3).

The brain of *Megalagus* was almost entirely lissencephalic, with the exception of weakly invaginated lateral sulci that extend the length of the cerebrum just lateral to the sagittal sinus. The anterior and posterior extremities of the lateral

sulci diverge laterally, while in the mid-length the sulci approach the superior sagittal sinus (figure 3). The rhinal fissure, marking the division between the palaeocortex and neocortex [27,28], is visible in *Megalagus* as a weak sulcus. It runs longitudinally on both sides of the endocast (the right side, being damaged, does not show its whole course; figure 3). There is no apparent Sylvian sulcus in *Megalagus*. The superior sagittal sinus is visible in the caudal half of the endocast of *Megalagus*, whereas in the rostral half there is a superior sagittal sulcus, but the sinus is not apparent, which suggests that it would have been partially buried in the meninges. In extant lagomorphs, the superior sagittal sinus is continuous with the transverse sinus leading to the sigmoid sinus, which is continuous with the internal jugular vein. In *Megalagus*, the transverse sinus is strong and well developed, but the area where it would be expected to merge with the sigmoid sinus is obscured by a large, elongate temporal foramen (as described in [25], p. 509), located between the squamosal, parietal and the mastoid exposure of the petrosal (figure 2). A temporal foramen this large is unknown for any extant lagomorph. On the endocast, the

**4**

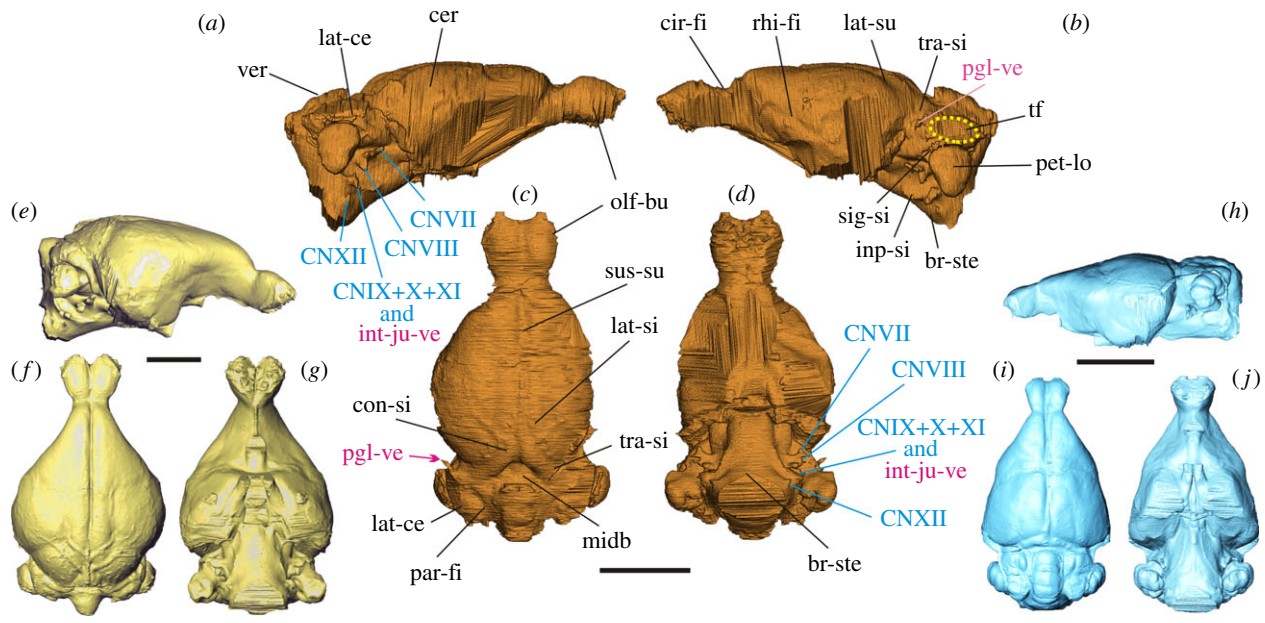

**Figure 3.** Digital endocasts of *Megalagus turgidus* (*a–d*), extant leporid *Poelagus marjorita* AMNH 51052 (*e–g*), and extant ochotonid, *Ochotona princeps* AMNH 40547 (*h–j*). Abbreviations: br-ste, brain stem; cer, cerebrum; cir-fi, circular fissure; CNVII, cranial nerve VII (facial nerve); CNVIII, cranial nerve VIII (vestibulocochlear nerve); CNIX, cranial nerve IX (glossopharyngeal nerve); CNX, cranial nerve X (vagus nerve); CNXI, cranial nerve XI (accessory nerve); CNXII, cranial nerve XII (hypoglossal nerve); con-si, confluence of sinuses; inp-si, inferior petrosal sinus; int-ju-ve, internal jugular vein; lat-ce, lateral lobe of cerebellum; lat-si, lateral sinus; lat-su, lateral sulcus; midb, midbrain; olf-bu, olfactory bulbs; par-fi, paramedian fissure; pet-lo, petrosal lobule; pgl-ve, postglenoid vein; rhi-fi, rhinal fissure; sig-si, sigmoid sinus; sus-su, superior sagittal sulcus; tf, temporal foramen; tra-si, transverse sinus; ver, vermis. Colour code: blue, nerves; pink, blood vessels; black, brain structures; (*a,b,e* and *h*), lateral; (*c,f* and *i*) dorsal; (*d,g* and *j*), ventral views. Surface rendering of *Megalagus* endocast available in [26]. (Online version in colour.)

temporal foramina are apparent as large flat areas on top of the lateral lobes of the cerebellum (figure 3).

The cerebrum does not cover the cerebellum in *Megalagus turgidus* (figure 3); thus, a portion of the midbrain of *Megalagus* is exposed. The anteroposterior extension of the midbrain in *Megalagus* is less expansive than observed in the basal Glires *Rhombomylus* (see [5]: figure 50), which has a very broadly exposed midbrain. There are no clearly defined colliculi. The dorsal surface of the cerebellum exhibits a well-defined vermis that is separated from the lateral lobes by the paramedian fissures (figure 3). The lateral lobes of the cerebellum are large, although their precise form is somewhat obscured by the impression of the temporal foramen on the endocast. The cerebellar part of the endocast in *Megalagus* is positioned slightly ventral to the cerebral hemispheres (figure 3). This configuration is caused by the arching of the skull, which is expressed strongly both in leporids [29] and in *Megalagus* (figure 2), and is much weaker in *Ochotona* [6]. The petrosal lobules (comprised the paraflocculi) are large, with a rounded and smooth, teardrop-shaped (tapering ventrally) lateral surface. They are aligned with the lateral lobes and the vermis in dorsal view. Excluding the petrosal lobules, the cerebellum of *Megalagus* is slightly narrower than the cerebrum (table 1).

The ventral aspect of the cranium is damaged, which does not allow for the reconstruction of some of the more rostral structures on the endocast (e.g. the optic tracts). The hypophyseal fossa is not clearly demarcated, with the relevant region actually being concave rather than convex on the endocast (figure 3).

The brainstem is well preserved (slightly curved ventrally), thus the more posterior vascular foramina and casts of some cranial nerves can be reconstructed. The casts of the passageways for cranial nerves VII (facial) and VIII (vestibulocochlear) are visible on both sides, located rostral to the petrosal lobules (figure 3). Similarly, a cast of the jugular foramen, through which the internal jugular vein and the IX (glossopharyngeal), X (vagus) and XI (accessory) cranial nerves would have passed, is positioned rostroventral to the petrosal lobule (figure 3). Finally, there is a cast of the hypoglossal foramen for cranial nerve XII (hypoglossal) on the brainstem, expressed better on the left side of the endocast (figure 3).

## 3. Comparisons with extant and fossil Lagomorpha

Based on comparisons to endocasts of extant members of Lagomorpha (electronic supplementary material, figure S3), the endocast morphology of *Megalagus* can be interpreted to exhibit an array of primitive and derived characters, the latter mostly in a nascent form, representing features developed further in crown lagomorphs. The primitive characters displayed by *Megalagus* include large olfactory bulbs that are separated from the cerebrum by a wide circular fissure, partly exposed midbrain, and generally much less expanded cerebral hemispheres, in comparison with recent lagomorphs and archaeolagines (see the electronic supplementary material, figure S3, [20]).

Compared to extant lagomorphs, the endocast of *Megalagus* (especially in dorsal and ventral views) shows a greater similarity to leporids (and to exclusively fossil archaeolagines) than to ochotonids. As in leporids, the cerebral hemispheres are round rather than triangular in outline, which contrasts with the shape of the hemispheres in extant ochotonids (figure 3). Additionally, the *Megalagus* endocast shows a slight downward bending (mostly at the brain stem), reflecting the arched profile of the cranium, also

**Table 1.** Measurements and parameters of the endocast of *Megalagus turgidus* (FMNH UC 1642). (Calculations of encephalization quotient (EQ) and masses of olfactory bulbs and petrosal lobules based on an estimated body mass of *Me. turgidus* of 2325 g; reconstructed data for NS and NS/TS in italics.)

| measurement (abbreviation); values in mm | |
| --- | --- |
| total length (TL) | 37.76 |
| olfactory bulb length (OL) | 8.32 |
| olfactory bulb width (OW) | 9.60 |
| olfactory bulb height (OH) | 5.61 |
| neocortex maximum height (NMH) | 10.21 |
| cerebrum total length (CRML) | 19.20 |
| cerebrum maximum width (CRMW) | 18.24 |
| cerebrum maximum height (CRMH) | 12.80 |
| cerebellum length (vermis) (CLML) | 9.28 |
| cerebellum width (without petrosal lobules) (CLW) | 16.32 |
| **ratio; values in %** | |
| OL/TL | 22.03 |
| CRML/TL | 50.85 |
| CLML/TL | 24.58 |
| CLW/CRMW | 89.47 |
| OW/CRMW | 52.63 |
| OW/CLW | 58.82 |
| NMH/CRMH | 79.77 |
| **surfaces (abbreviation); values in mm$^2$** | |
| total endocast area (TS) | 4108.10 |
| neocortical surface area (NS) | *779.86* |
| neocortical surface area (one side) (NS1) | 389.93 |
| **volumes (abbreviation); values in mm$^3$** | |
| total endocast (TV) | 7052.78 |
| olfactory bulbs (OV) | 280.10 |
| petrosal lobules (PLV) | 162.59 |
| **ratio; values in %** | |
| NS/TS | *18.98* |
| OV/TV | 3.97 |
| PLV/TV | 2.31 |
| **mass; values in mg** | |
| olfactory bulb mass | 266.76 |
| petrosal lobule mass | 154.85 |
| **Jerison's EQ** | 0.31 |
| **Eisenberg's EQ** | 0.39 |

characteristic of leporids (figure 2; electronic supplementary material, figure S4). In lateral view, however, the *Megalagus* endocast exhibits a much flatter (less swollen) cerebrum, which resembles more closely that of *Ochotona* (figure 3; electronic supplementary material, figure S3I–K). The endocast of *Megalagus* is consistent with modern lagomorph taxa in its location in the cranium, with the maximal width being at the level of the posterior root of the zygomatic arch (see [20]). The location of the anterior edge of the olfactory bulbs matches that observed in extant leporids (above P4–M2 alveolus; electronic supplementary material, figure S4A), but it is more posterior than in modern *Ochotona* (above P3; electronic supplementary material, figure S4B).

The olfactory bulbs of *Megalagus* are larger in terms of their volume relative to the endocranial volume and their width relative to the cerebrum maximal width than those of all extant lagomorphs for which data are currently available (figure 4*c*; electronic supplementary material, figure S5; table 1; electronic supplementary material, table S2). On the other hand, the volume of the olfactory bulbs relates to the estimated body mass groups *Megalagus* with extant leporids (electronic supplementary material, figure S5). In terms of the length of the olfactory bulbs relative to the endocast length (OL/TL, table 1), *Megalagus* overlaps with some of the *Lepus* species (i.e. *Lepus americanus*, electronic supplementary material table S2) and it is close to the archaeolagine *Hypolagus brachygnathus* and fossil ochotonid *Prolagus meyeri*, for which the relative length of the olfactory bulbs is *ca.* 20% [19]. Among the extant lagomorphs, only the leporid *Poelagus marjorita* markedly exceeds *Megalagus* in this respect, whereas the rest of the studied taxa exhibit lower values for the OL/TL ratio (table 1; electronic supplementary material, table S2). Thus, such an elongation of the olfactory bulbs seems to be generally typical of major lagomorph lineages, including *Megalagus*, Prolaginae and extant leporids.

The cerebral hemispheres of *Megalagus* are less laterally expanded than in modern lagomorphs and archaeolagines, thus its endocast lacks the overall teardrop shape displayed by the crown taxa (figure 3; electronic supplementary material, figure S3). In *Megalagus*, the circular fissure is broader than in ochotonids, and much broader than in leporids, in which the cerebrum almost touches the olfactory bulbs owing to the swollen cerebral hemispheres (figure 3; electronic supplementary material, figure S3).

The endocasts in all lagomorph taxa studied here are lissencephalic (smooth). The lateral sulci are only weakly present on the *Megalagus* endocast (figure 3; electronic supplementary material, figure S2). The rhinal fissure in the rostral half of the cerebrum in *Megalagus* is similarly positioned to that of modern lagomorphs, but in the temporal region its course is more dorsal than in living leporids (electronic supplementary material, figure S3), indicating much lesser neocorticalization, as also reflected in the relatively lower neocortical ratio (figure 4; electronic supplementary material, table S2).

In *Megalagus*, the midbrain is more broadly exposed than in extant lagomorphs, although the difference is slight. Overall, extant leporids have less exposed midbrain than *Ochotona*, probably owing to the characteristic arched form of the leporid cranium, and to greater development of the transverse sinus, which partly covers the midbrain (electronic supplementary material, figure S3). The colliculi are variably visible in extant lagomorph endocasts (electronic supplementary material, figure S3). In *Megalagus*, they are indiscernible (figure 3), similar to those in most leporids, whereas the caudal colliculi are more apparent in *Ochotona* (electronic supplementary material, figure S3).

The cerebellum in modern lagomorphs (both leporids and ochotonids) is markedly narrower than the cerebrum (electronic supplementary material, figure S3, table S2). By contrast, in *Megalagus* the cerebrum and cerebellum are of similar width (table 1). The petrosal lobules are very well developed in all lagomorphs, including *Megalagus* (figures 2

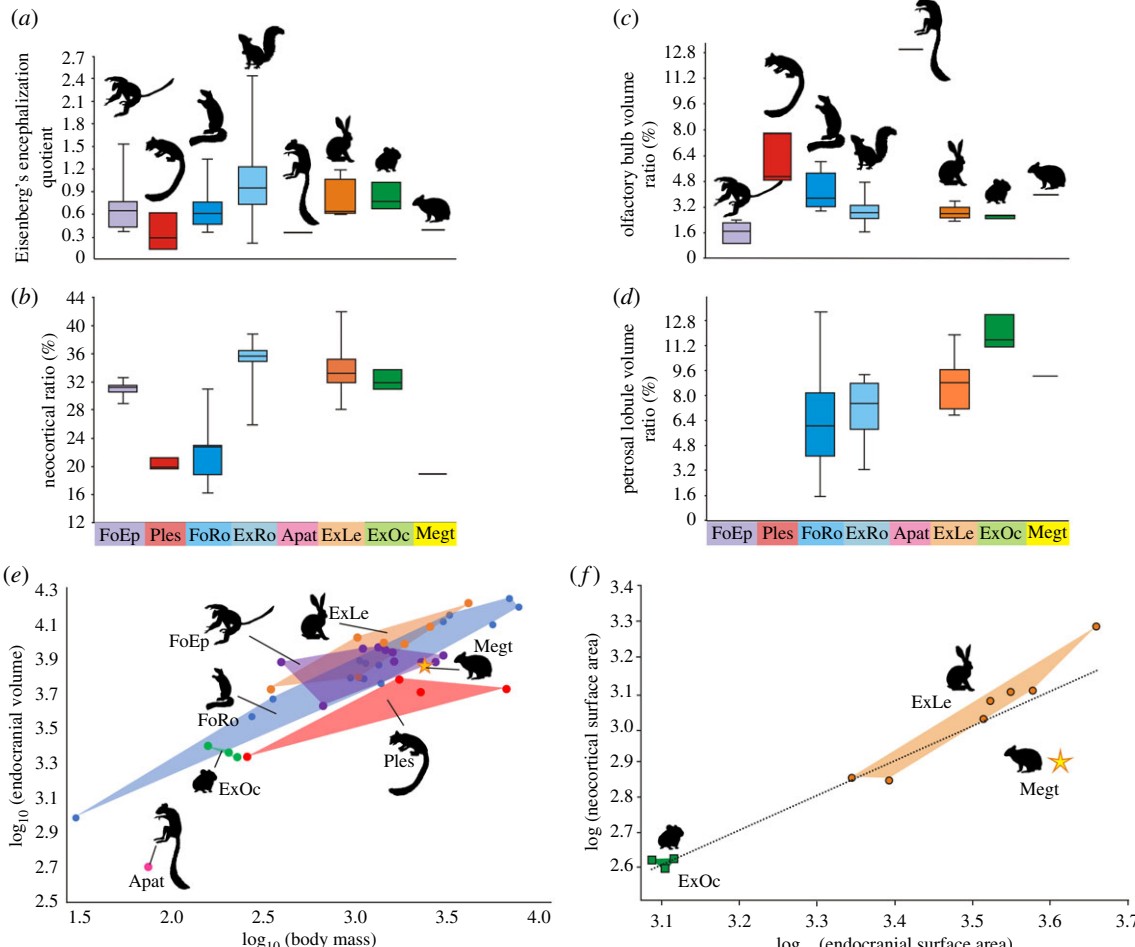

**Figure 4.** Comparisons of endocranial metrics in Euarchontoglires. (*a*) Encephalization quotient (EQ) based on Eisenberg's [30] equation; (*b*) neocortical surface area ratio (NS/ESx100); (*c*) olfactory bulb volume ratio (OV/TVx100); (*d*) petrosal lobule volume ratio (PV/TVx100); (*e*) bivariate plot of log$_{10}$ (endocranial volume) versus log$_{10}$ (body mass); and (*f*) bivariate plot of log$_{10}$ (neocortical surface area) versus log$_{10}$ (endocranial surface area). Abbreviations: Apat, Apatemyidae; ExLe, extant leporids; ExO, extant ochotonids; ExRo, extant rodents; FoEp, fossil euprimates; Megt, *Megalagus turgidus*; Ples, Plesiadapiformes. (Online version in colour.)

and 3, electronic supplementary material, figure S3). The volume of the petrosal lobules relative to that of the whole endocast in *Megalagus* is within the range of values known for extant leporids (figure 4; electronic supplementary material, figure S5), although it is still lower than in extant ochotonids (figure 4*d*). On the other hand, the volume of the petrosal lobules in relation to the estimated body mass is lower in *Megalagus* than in any modern lagomorph (electronic supplementary material, figure S6A).

## 4. Comparisons with other Euarchontoglires

The morphology of the *Megalagus* endocast is similar in many ways to that observed in early members of Euarchontoglires (electronic supplementary material, figure S7), such as Plesiadapiformes [31–34], Ischyromyidae (e.g. see [35–37]) and *Rhombomylus*, a eurymylid [5]. These similarities include well-defined, uncovered and elongated olfactory bulbs, partly exposed midbrain and prominent petrosal lobules.

The endocast volume relative to the estimated body mass for *Megalagus* places it on the margin of the distribution of fossil euprimates and close to ischyromyid rodents and plesiadapiforms, but separate from the extant lagomorphs (figure 4). The relative length of the olfactory bulbs (OL/TL) in *Megalagus* is higher than in most of the ischyromyid rodents (apart from a borderline case of *Paramys copei*) [37] and early representatives of modern rodent groups, such as sciurids or

aplodontids) [38,39]. This ratio overlaps partly with the values observed for the Palaeocene–Eocene Plesiadapiformes (electronic supplementary material, table S4) [31–33] and early Eocene eurymylid *Rhombomylus* (*ca* 22%). The olfactory bulb relative volume (OV/TV ratio) gives a slightly different picture, with the value for *Megalagus* (table 1) being within the range of ischyromyid rodents (figure 4*c*) [39]; this contrast reflects some difference in the shape of the bulbs, with them being more elongate relative to their volume compared to early fossil rodents. The OV/TV ratio value is higher than in early representatives of the modern crown rodents [37–39], but slightly lower than in Plesiadapiformes (electronic supplementary material, table S3) [31,32].

The relative length of the cerebral hemispheres (CRML/TL) is comparable in *Megalagus* (51%; table 1), Ischyromyidae and Plesiadapiformes (electronic supplementary material, table S4), the latter showing the greatest observed range, with *Plesiadapis* showing the lowest values (42–45.5%) and *Microsyops* displaying proportionally the longest cerebrum (54.0%). This primitive brain architecture shared by *Megalagus* with Ischyromyidae and Plesiadapiformes is in contrast with more derived modern lineages of Glires [39] and euprimates [40], which all exhibit greater relative cerebral development (the CRML/TL ratio on average is over 60%).

Another feature shared by *Megalagus*, ischyromyid rodents [35,37], Plesiadapiformes [31,32] and early members of the modern rodent clades [38,39] is an exposed midbrain.

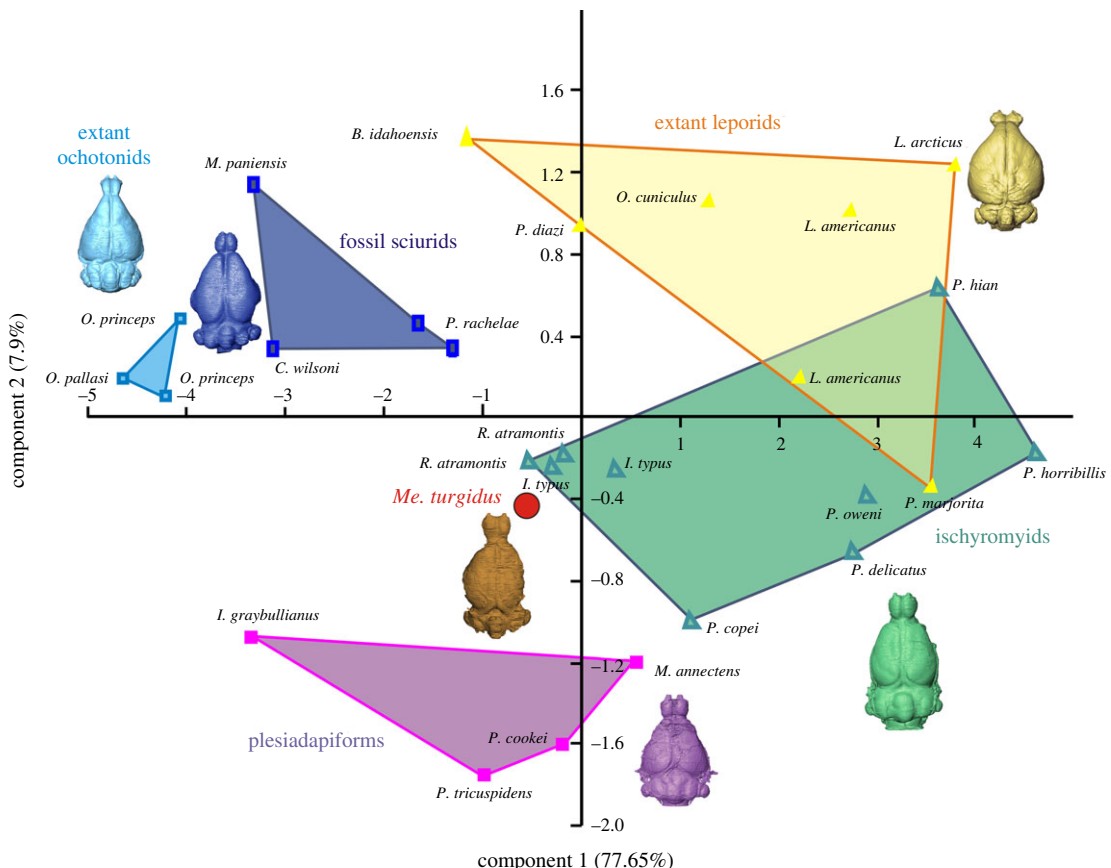

**Figure 5.** Results of principal component analysis of nine endocast characters in 24 species of fossil and extant lagomorphs, ischyromyids, fossil sciurids and plesiadapiforms. Note the central position of *Megalagus* in the morphospace, closer to Ischyromyidae and Plesiadapiformes rather than to extant Lagomorpha. For raw data, see the electronic supplementary material. (Online version in colour.)

The extent of the midbrain exposure in *Megalagus* is much less than in *Plesiadapis tricuspidens* [33] and *Rhombomylus* [5], as well as in some of the Ischyromyidae (*Paramys*, *Noto-paramys* and *Ischyromys*) [35,37], in which the colliculi are visible to some degree. It resembles more closely that of the virtual endocast of *Microsyops annectens*, in which the colliculi are not visible, but a patch of midbrain is exposed, and the rest of the midbrain is grooved by the venous sinuses rather than the cerebrum [32] (also see [41]). However, it is worth noting that exposure of the colliculi is known to vary intraspecifically (e.g. in *Mi. annectens* and *Ischyromys typus*) [32,36].

The cerebellum of *Megalagus* is only marginally narrower than the cerebrum, a common trait among early fossil Euarchontoglires, excluding fossil euprimates (e.g. see [33,34,39,42]). The difference between the cerebrum and cerebellum width is greater in more derived fossil taxa (e.g. *Cedromus wilsoni*) [38], in extant representatives of modern Glires (compare figure 3; electronic supplementary material, figure S3), and in living and fossil euprimates [40]. The vermis and cerebellar hemispheres are generally similarly developed in *Megalagus* as in Ischyromyidae and Plesiadapiformes, but the relative size of the petrosal lobules is distinctly larger in *Megalagus* (figure 4d; electronic supplementary material, table S3) than in extant and fossil rodents.

## 5. Principal component analysis

A principal component analysis was performed based on nine measurements of the endocasts for 24 species of extant lagomorphs, fossil rodents, plesiadapiforms and *Megalagus*

(electronic supplementary material, table S4). Principal component 1 and 2 (PC1 and PC2) represent 77.65% and 7.9% of the variance, respectively (electronic supplementary material, figure S8), whereas principal components 3–5 represent a further 12% of the variance (electronic supplementary material, figure S8). All the variables are positively correlated with PC1 (electronic supplementary material, figure S8), which implies that it represents a proxy for the size of the endocast. PC2 is most strongly correlated with the olfactory bulb length (−0.57) and width (0.58). Generally, an endocast with a high PC2 score has relatively short but wide olfactory bulbs (electronic supplementary material, table S4).

In the plot of PC1 versus PC2, *Megalagus* falls between smaller ischyromyids (*Ischyromys*) and plesiadapiforms (figure 5). Extant leporids and larger ischyromyid rodents partly overlap in the morphospace, while extant ochotonids are grouped more closely with the representatives of fossil sciurids (figure 5). *Megalagus* is relatively separated from extant lagomorphs (both Ochotonidae and Leporidae, although closer to the latter) and sciurids, which points to its morphological difference from the modern groups of Glires. Overall, *Megalagus* plots near the centre of the morphospace of the studied Euarchontoglires, and this equidistant position suggests its morphological consistency with the endocast architecture characteristic of basal groups of Euarchontoglires.

## 6. Discussion

The endocranial morphology of *Megalagus* can be interpreted from several perspectives; first, it represents an early stage of

brain evolution in lagomorphs, and second, more broadly, a source of information relevant to understanding the form of the brain in Euarchontoglires in general.

Compared to extant lagomorphs and rodents, including early fossil representatives of modern crown rodents (e.g. Oligocene *Cedromus*) [38], the encephalization quotients (EQ) of *Megalagus* and the rest of the sampled basal members of Euarchontoglires are generally lower (figure 4a; electronic supplementary material, table S3). The neocortical ratio shows similar values for *Megalagus* (19%), ischyromyid rodents (17–23%) and plesiadapiforms (20–24%), all of which display smaller relative area of the neocortex (figure 4; table 1; electronic supplementary material, tables S2, S3) than in the crown members of Glires, and in both fossil and extant members of Euprimates, in which the ratio value is generally over 30% ([38–40], figure 13). These observations suggest that expansion of the brain through neocorticalization happened multiple times in the evolution of Euarchontoglires, and that the common ancestor of this group probably had a relatively small neocortex [33]. With respect to the olfactory bulbs, *Megalagus* is similar to plesiadapiforms and early fossil rodents in their size (length) relative to the rest of the brain (electronic supplementary material, table S4), with relatively larger bulbs than observed in living lagomorphs, or fossil or living euprimates (figure 4c). This similarity suggests that, in this way, *Megalagus* is also similar to what might be inferred for the primitive ancestor of Euarchontoglires. The one issue with that interpretation is the relatively much larger olfactory bulbs of apatemyids. This part of the endocast in *Labidolemur* and *Carcinella* (see [42] and [43], respectively) match in proportions and overall shape the olfactory bulbs of the late Cretaceous eutherians (e.g. *Asioryctes*) [44]. Whether this similarity is a matter of ancestor–descendent relationship or convergent evolution remains uncertain. It is also worth noting that the contrast with living lagomorphs is not apparent when olfactory bulb size is considered in relation to body mass rather than endocranial volume (electronic supplementary material, figure S5), which suggests that the apparent difference in relative olfactory bulb size might be attributable to expansions in other parts of the brain rather than reduction in the apparatus for olfaction.

In *Megalagus*, the anterior extremities of its olfactory bulbs reach the area over M1; thus, they are more posteriorly located in the cranium compared to modern lagomorphs (up to P3–P4; electronic supplementary material, figure S4) and rodents (up to the diastema) [39]. With respect to the position of the olfactory bulbs, *Megalagus* is similar to the middle-to-late Eocene ischyromyids (*Rapamys* and *Ischyromys*) [35,36], and to the early Eocene paromomyid plesiadapiform *Ignacius* [31], and so might be primitive in this way relative to modern lagomorphs. In ischyromyids, stratigraphically younger genera have more anteriorly

located olfactory bulbs, but the bulbs extend no farther than M1, while in the fossil representatives of the modern rodents the olfactory bulbs are positioned rather at P4 (e.g. *Cedromus*) [38]. Silcox *et al*. [32] suggested that the relative position of the olfactory bulbs in plesiadapiforms may be related to the length of the muzzle, especially the diastema. However, in early Glires (both ischyromyids, and lagomorphs), the diastema length does not change significantly; thus, the explanation for the shifting relative position of the front of the brain has to be different.

In all Palaeogene taxa studied herein the midbrain is exposed, although to varying degrees. The relatively well-exposed midbrain, which is partially visible between the posteriorly diverging and relatively long cerebral hemispheres, was characteristic of stem placentals, such as *Asioryctes*, *Barunlestes*, *Kennalestes* and *Zalambdalestes* [44,45]. As such, this trait of *Megalagus* is likely to be an archaic (plesiomorphic) feature, comprising part of the morphotype of early Euarchontoglires.

In summary, the stem groups of Euarchontoglires share features that include relatively large olfactory bulbs, an at least partially exposed midbrain, the cerebellum with relatively large petrosal lobes, low EQ and a small neocortex. The similarities in taxa that range from the late early Palaeocene *Plesiadapis* to the early Oligocene *Megalagus* suggest that this general architecture remained relatively stable, especially in Lagomorpha. In that context, the endocast structure of *Me. turgidus* fills the gap in our knowledge of the primitive brain morphology for Lagomorpha, arguably the most basal branch of Glires, and as such it gives us an insight into the ancestral brain architecture of Euarchontoglires.

**Data accessibility.** Data are available from the Dryad Digital Repository: https://doi.org/10.5061/dryad.0vt4b8gwg [26].

**Authors' contributions.** Ł.F.F. conceived the study; Ł.F.F. and M.T.S. designed and outlined research; all authors studied specimens, gathered and analysed data; S.L.T., Ł.F.F and M.T.S. wrote the paper, all authors discussed and approved the manuscript.

**Competing interests.** We declare we have no competing interests.

**Funding.** The present work was supported by a Marie Skłodowska-Curie Actions: Individual Fellowship (H2020-MSCA-IF-2018-2020; grant no. 792611) to O.C.B., a Natural Sciences and Engineering Research Council of Canada (NSERC) CGS grant to M.M.L., an NSERC Discovery Grant to M.T.S., and a National Science Centre (Cracow, Poland) grant no. 2015/18/E/NZ8/00637 to Ł.F.F.

**Acknowledgements.** We are grateful to W. Simpson (FMNH) for access to the *Megalagus* specimen, E. Westwig (formerly AMNH) for making *Romerolagus diazi* available for study, J.O. Thostenson (formerly AMNH) for CT-scanning of the AMNH and FMNH lagomorph specimens at the AMNH, and D. Boyer for facilitating the scanning at the Shared Materials Instrumentation Facility (SMIF), Duke University. We thank the artists A. Kapuścińska who made a reconstruction of the *Megalagus* head and F. Ippolito for the photos of the skull. Thanks are due to three anonymous reviewers, whose comments substantively improved this paper.

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
