## [Reviewer comments · Proceedings of the Royal Society B: Biological Sciences]

Review History

RSPB-2020-0665.R0 (Original submission)

Review form: Reviewer 1

Recommendation

Accept with minor revision (please list in comments)

Scientific importance: Is the manuscript an original and important contribution to its field?

Excellent

General interest: Is the paper of sufficient general interest?

Excellent

Quality of the paper: Is the overall quality of the paper suitable?

Excellent

Is the length of the paper justified?

Yes

Should the paper be seen by a specialist statistical reviewer?

No

Do you have any concerns about statistical analyses in this paper? If so, please specify them explicitly in your report.

No

It is a condition of publication that authors make their supporting data, code and materials available - either as supplementary material or hosted in an external repository. Please rate, if applicable, the supporting data on the following criteria.

Is it accessible?

Yes

Is it clear?

Yes

Is it adequate?

Yes

Do you have any ethical concerns with this paper?

No

Comments to the Author

The study of the morphology of the *Megalagus* endocast reveals a structure similar in many ways to that observed in early members of Euarchontoglires. The endocast of *Megalagus* may therefore provide crucial information on the condition of the brain near the split between basalmost Glires and Euarchonta. These similarities include well-defined, uncovered, and elongated olfactory bulbs, partly exposed midbrain, and prominent petrosal lobules. The two later parameters call for further remarks. See below

Abstract : « Primitive aspects of the endocranial morphology in *Megalagus* include ... exposure of the midbrain »,

also page 7, line 153-154

page 9, line 194

page 10, line 214 and 237-238

page 14, line 317-322

Midbrain exposure is not a primitive (archaic is I think a better word) character. following Starck, D. 1963. "Freiliegendes Tectum mesencephali" ein Kennzeichen des primitiven Säugetiergehirns? *Zoologischer Anzeiger* 171: 350-359. It would be worth checking this publication, is it outdated or not taking into account recent and current studies ?

See also Kass, J.H., Collins, C.E. 2001. Variability in the sizes of brain parts. *Behavioral and Brain Sciences* 24(2) : 288-290.

Please look at this note especially concerning page 9, line 197-198 : « The colliculi are variably visible in extant lagomorph endocasts » visible or exposed ? and page 11, line 245-246 « exposure of the colliculi is known to vary intraspecifically (e.g., in *Microsypops annectens* and *Ischyromys typus*) (30, 34). »

Page 5, line 92: Is « the jugal arches » the correct term ? or zygomatic arch ?

Page 6, line 119: the temporal foramina are not indicated on figure 3 (should be tf) but on figure 2

Page 6, line 133-134 : « Excluding the petrosal lobules, the cerebellum of *Megalagus* is slightly narrower than the cerebrum »,

also page 9, line 203

page 11, line 253

About the volume of the petrosal lobules, be careful that the cast of the cerebellum is probably a

good proxy for cerebellum shape and volume but in the case of the petrosal lobules of the paraflocculus, the putative proxy is the subarcuate fossa cast and subarcuate fossa size is not a reliable proxy for paraflocculus size :

« The correlation between the size of the petrosal lobule of the paraflocculus and the subarcuate fossa in adults is weak » (Sanchez-Villagra, M.R. 2002. The cerebellar paraflocculus and the subarcuate fossa in *Monodelphis domestica* and other marsupial mammals – ontogeny and phylogeny of a brain-skull interaction. *Acta Theriologica* 47(1) : 1-14)

It's worth checking whether this bony structure is endochondral, in which case the lack of strict relationship between the fossa and the nervous structure inside it simply results from the fact that the bone doesn't result from the ossification of a mesenchymal envelope covering the nervous structure (as it is the case for instance for the bones constituting the braincase)

Of interest concerning page 11, lines 247-254 and especially 251-252 : « The vermis and cerebellar hemispheres are generally similarly developed in *Megalagus* as in *Ischyromyidae* and *Plesiadapiformes* », the expansion of the cerebellar hemispheres appears to be related to the expansion of specific regions of the cerebral cortex ; see Smaers et al. 2018
<https://doi.org/10.7554/eLife.35696.001>

Review form: Reviewer 2

Recommendation

Major revision is needed (please make suggestions in comments)

Scientific importance: Is the manuscript an original and important contribution to its field?

Good

General interest: Is the paper of sufficient general interest?

Good

Quality of the paper: Is the overall quality of the paper suitable?

Acceptable

Is the length of the paper justified?

Yes

Should the paper be seen by a specialist statistical reviewer?

No

Do you have any concerns about statistical analyses in this paper? If so, please specify them explicitly in your report.

Yes

It is a condition of publication that authors make their supporting data, code and materials available - either as supplementary material or hosted in an external repository. Please rate, if applicable, the supporting data on the following criteria.

Is it accessible?

Yes

Is it clear?

Yes

Is it adequate?

No

Do you have any ethical concerns with this paper?

No

Comments to the Author**Overall Impressions**

In this submission, López-Torres and colleagues contribute a description of the fossilized endocast of a remarkably complete early Oligocene lagomorph skull (*Megalagus*). They qualitatively and quantitatively compare the endocast of *Megalagus* to other extant and extinct lagomorphs, rodents, and primates. This study is straight-forward and succinct, and the importance of the fossil is clear.

The strengths of the study lie in the incredible temporal and systematic context of this well-preserved fossil, as well as the remarkable comparative dataset the authors have amassed. Endocasts of extant and fossil mammals are not trivial to reconstruct from CT data, and the authors have done meticulous work building their digital dataset.

There are several book-keeping details that the authors should attend to before the manuscript can be accepted for publication, some of which may require major revisions. In this regard, I have been a bit more detail-oriented than usual due to the global pandemic. My apologies if this review comes off as too pedantic—it is my intention to help ensure the highest-quality end product to match the effort that the authors have clearly put in.

Book-Keeping and conceptual issues to address throughout:

1. Sample sizes are never directly reported, and key evolutionary conclusions are drawn from boxplots of skewed sample sizes (e.g. 21 extant rodents vs. 7 extant lagomorphs, judging by Table S4a). More even sample sizes between extant groups (for which obtaining additional specimens is more feasible) would ensure robusticity of the results.
2. Many of the key figures that are central to the results and discussion are only present in the supplemental information. For example, Figures S1, S2, and S4 contain visual information that is crucial to interpretation of the comparisons. Presumably it is the comparative evolutionary morphology that makes the description of this endocast suitable for a broad audience such as that of Proceedings B. By including only figures of the *Megalagus* endocast in the body of the manuscript, the authors have displaced evidence of their descriptive arguments as well as the bulk of their work. I strongly encourage the authors to identify a format and/or make revisions to the figures that would allow more of the total body of morphological evidence to be included in the main body of the text.
3. Why is *Megalagus* compared with both extant and extinct Glires, but only fossil Euprimates? Numerous evolutionary arguments are made about the directionality of brain region shifts in the context of primate evolution, but no extant primate endocasts are quantified.
4. Why are different taxonomic samples used for the boxplots, biplots, and PCA? E.g. extant rodents are absent from the biplot and PCA, but the reason this exclusion is never explained.

Specific Manuscript Sections:**Introduction:**

Introduction is succinct and provides the necessary systematic and temporal context needed to demonstrate the import of the *Megalagus* endocast to interpretation of Euarchontoglires evolution. Figure 1 (phylogenetic scheme) is helpful and may be improved if revised as a phylogeny that annotates primitive and derived traits (see “Comparison with extant and fossil Lagomorpha” section below).

Description

A reference to a nomenclatural convention would enhance the reader's ability to evaluate the appropriateness of structure identification in the description. This is especially critical in comparative neuroanatomy, where terms coined for human brain anatomy have been variably applied in zoological contexts. A standardized ontology would allow the reader to follow designations based on structural associations, homology, or both. The *Nomina Anatomica Veterinaria* and Swanson's Brain Maps come to mind as having helpful ontologies, and the authors have an extensive body of previous work and endocast evolution reviews they could refer to as well.

Throughout: Please check to ensure that figure references are placed after the described morphology rather than before, e.g. Line 96: "The endocast is elongate, with well-developed (Fig. 3), ovoid, and pedunculated olfactory bulbs (Fig. 3)..."

Line 108: From the figure provided (Fig. 3), I agree with the authors that there is no apparent Sylvian sulcus in *Megalagus*; however, I am not sure I confidently agree with their interpretation of the rostral extent of the rhinal fissure as a Sylvian fossa. Providing a reference definition for the Sylvian fissure/fossa based on e.g. structural associations would ensure agreement.

Lines 110–115: Clarification is needed regarding superior sagittal sinus/sulcus morphology: In the discussion, the authors point out that "The superior sagittal sinus is visible in the caudal half of the endocast of *Megalagus*, whereas in the rostral half there is a superior sagittal sulcus, but the sinus is not apparent..." However, "superior sagittal sinus" is labeled rostrally in Figure 3, near the olfactory bulbs, contradicting the text.

Comparisons with extant and fossil Lagomorpha

I understand that Proceedings B has significant page restrictions and financial penalties for going over a 6-page limit; however, the authors have an extremely helpful (and beautiful!) comparative endocast figure in their supplemental document. I would strongly suggest trying to find a way to include comparative figure S1 in the main body of the text.

The discussion of primitive characters displayed by *Megalagus* is a vital component to the manuscript—inclusion of the comparative figure S1 into the main body of text would greatly enhance the interpretability of the manuscript. In addition to including the anatomical figure, highlighting the traits discussed on a phylogeny would help readers interpret similarities between *Megalagus* and, for example, leporids vs. ochotonids.

Line 187: "The brain in all lagomorph taxa studied here is lissencephalic": should "brain" read "endocast" as the brain in *Megalagus*, a lagomorph taxon, is unavailable?

Discussion

Lines 290–292: "With respect to the olfactory bulbs, *Megalagus* is similar to plesiadapiforms and early fossil rodents in their size relative to the rest of the brain, with relatively larger bulbs than observed in living lagomorphs, or fossil or living euprimates (Fig. 4C)."

This is an interesting pattern, especially in light of recent literature documenting tradeoffs in olfactory morphology with cerebral hemisphere volume and/or thermoregulatory structures in mammals and birds; however, I am concerned that the observed pattern may be an artifact of sample size. In figure 4C, the value for *Megalagus* olfactory bulb volume ratio only falls within the box plot ranges for fossil and extant rodents. All of the box plots that represent groupings within Glires appear to be relatively conserved, and the patterns of evolutionary brain region shifts the authors are discussing may be driven by an imbalance in sample size rather than a real evolutionary pattern. From Table S4a, there are 300% more extant rodents than extant

lagomorphs (note that I was unable to find any direct reporting of sample sizes in the main text or supporting information and counted from the table). The low amount of variance in lagomorphs and ochotonids compared to the rodents may be driven by the low sample size (N= 7 & 3 respectively). The authors should include additional lagomorph samples to ensure that their conclusions are robust to sample size equalization. Finally, comparisons with pleasiadapiforms may be over-stated if this within-Glires pattern fails to hold up to additional specimens.

Lines 304–306: “In *Megalagus* the anterior extremities of its olfactory bulbs reach the area over M1; thus, they are more posteriorly located in the cranium compared to modern lagomorphs (up to P3–P4; Fig. S2).” Parenthetical needs reference to Fig. 2i so comparison can be visualized.

Tables, Figures, & Captions:

Table 1: The measurement table is extremely helpful, and the tables in the supplemental provide a wealth of information. However, the measurements are not illustrated on a specimen, nor are anatomical landmark criteria for the measurements provided. Minimally, such a figure is necessary in the supplemental, and ideally the authors might also include landmark criteria (either written into the text or referred to a source).

Comparing Table 1 with Table S3: Why was olfactory bulb height measured in *Megalagus* but omitted from the PCA measurements?

Figure 3:

Please list structure abbreviations in alphabetical order and ensure that all structures are divided by consistent punctuation. Color coding of labels should be explained.

Caption is missing abbreviation for: “cer,” “CN XII,” “lat-si,” “midb,” “olf-bu,” “Sy-fo”

The structures “lat-si” in dorsal view and “lat-su” in lateral view appear to be the same structure. A single naming convention should be adhered to in the figure and caption.

Figure 4:

Both fossil and extant rodents are included in the box plots, but extant rodents do not appear to be included in the bivariate plot. These specimens should be included or a reason should be provided for why they are not. The sample size for the number of specimens in each box plot is not directly reported in the main text or supplemental.

Figure 5:

Why are extant rodents omitted from the PCA? The color-coding for groups in the PCA should be made consistent with the color coding in the box plots and biplot.

Supporting Information:

Materials & Methods:

Scanning and rendering methods for *Megalagus* are adequate. Scan parameters should also be provided for the comparative specimens, if even in the form of providing the morphosource links.

Fig. S1: Naming conventions are not standard throughout the figure caption (e.g. Leporidae is pointed but, but Ochotonidae is not; color-coding is not described). Anatomical labels are needed to satisfactorily compare other lagomorph endocasts with the *Megalagus* endocast.

Fig. S6: Anatomical labels are needed to satisfactorily compare Euarchontoglires endocasts with the *Megalagus* endocast.

Fig. S7: From the manuscript text, it is implied that the numerical features in the loading plots correspond with the order that the features are listed in Table S3 (i.e. loading factor 1 is endocast TL). Please include confirmation of this in the Fig S7 caption.

Table S4a: Caption needs correction: "Data used for the box plot analyses in Figure 4B, 4C, and 4C." In the Group column of the table, specify that Leporidae and Ochotonidae specimens are extant and Euprimates are extinct to maintain naming conventions of the main text.

Review form: Reviewer 3

Recommendation

Accept with minor revision (please list in comments)

Scientific importance: Is the manuscript an original and important contribution to its field?

Excellent

General interest: Is the paper of sufficient general interest?

Good

Quality of the paper: Is the overall quality of the paper suitable?

Good

Is the length of the paper justified?

Yes

Should the paper be seen by a specialist statistical reviewer?

No

Do you have any concerns about statistical analyses in this paper? If so, please specify them explicitly in your report.

No

It is a condition of publication that authors make their supporting data, code and materials available - either as supplementary material or hosted in an external repository. Please rate, if applicable, the supporting data on the following criteria.

Is it accessible?

Yes

Is it clear?

No

Is it adequate?

Yes

Do you have any ethical concerns with this paper?

No

Comments to the Author

The paper makes a significant contribution to the field of paleobiology by interpreting the neuroanatomy of a fossil that fills a critical gap in our knowledge of gliran and lagomorphan brain evolution. The authors do a good job of describing the condition that they observe in the fossil taxon. One area that could be improved is in contextualizing the fossil more by

summarizing major changes in neuroanatomy within Lagomorpha in an evolutionary and ecological context. While there is certainly a good discussion of similarities between taxa, what is missing is a broader context for those similarities. Similarly, while there are many functional arguments one could make from the neuroanatomy of this taxon, the authors seem hesitant to make them, which seems like an intentional oversight. If that is the intention, a reason should be given for that omission. Overall, I think the paper is important and worthy of publication. I think your figures are engaging, easy to follow, and informative. I look forward to seeing it published.

Line 72 – The authors state that the “the evolutionary history of the lagomorph brain is almost entirely unknown” making this study quite critical for addressing that knowledge gap however then raises the question of why they didn’t discuss evolutionary patterns along lagomorpha in the following text.

85 – For those not familiar with Glires and Euarchonta, it is not apparent why we should expect the basal condition to be different from the recent condition or indeed if the clades should be expected to differ at all. Why is it important to consider the primitive condition here? Are there major changes within Glires that this new specimen may inform the timing of?

102 – You may consider including a brief definition of lissencephalic

148 – I believe the authors should include a brief summary of trends among the comparative lagomorph sample, clearly defining what the primitive conditions in the clade are. It is difficult to follow the evolutionary context of the fossil specimen without general trends among lagomorphs clearly laid out.

Figure 2 is really quite lovely. You might consider rotating the lateral view of the endocast in the skull (i) slightly counter-clockwise so that it's at the same orientation as (c).

Within the Material and Methods, you should consider including a project number and DOI for surface files used in this project.

Fig. S3A, the Y-axis title is misspelled

Decision letter (RSPB-2020-0665.R0)

07-May-2020

Dear Dr Fostowicz-Frelik:

Your manuscript has now been peer reviewed and the reviews have been assessed by an Associate Editor. The reviewers’ comments (not including confidential comments to the Editor) and the comments from the Associate Editor are included at the end of this email for your reference. As you will see, the reviewers and the Editors have raised some concerns with your manuscript and we would like to invite you to revise your manuscript to address them.

To submit your revision please log into <http://mc.manuscriptcentral.com/prsb> and enter your Author Centre, where you will find your manuscript title listed under "Manuscripts with

Decisions." Under "Actions", click on "Create a Revision". Your manuscript number has been appended to denote a revision.

Research ethics:

Use of animals and field studies:

Please submit a copy of your revised paper within three weeks. If we do not hear from you within this time your manuscript will be rejected. If you are unable to meet this deadline please let us know as soon as possible, as we may be able to grant a short extension.

Best wishes,
Dr John Hutchinson, Editor
mailto:proceedingsb@royalsociety.org

Associate Editor
Board Member: 1
Comments to Author:
Dear authors,

Thank you for your submission to Proceedings B. Your submission has now been seen by three expert reviewers, all of whom find your work of substantial interest and high quality.

As one of the best explorations of early Euarchontoglires brain evolution, this contribution is bound to be very impactful.

As a testament to the thoroughness of your work, the referees have listed a number of suggested alterations that will clarify and improve a number of aspects of the paper, and their recommendations should be taken under consideration in a review.

Most significantly, Referee 2 suggests a very detailed review with a number of helpful considerations that should be addressed. Although I acknowledge that there are practical limitations such as page limits at play, I encourage the authors to take Ref. 2's recommendation for figure alterations under consideration; insofar as it is possible in light of page limits, it would be excellent to accommodate more of the comparative morphology that is currently in the supplement as this will be one of the most important lasting contributions from this work.

I look forward to viewing a revised version of this manuscript.

Reviewer(s)' Comments to Author:

Referee: 1

Comments to the Author(s)

The study of the morphology of the *Megalagus* endocast reveals a structure similar in many ways to that observed in early members of Euarchontoglires. The endocast of *Megalagus* may therefore provide crucial information on the condition of the brain near the split between basalmost Glires and Euarchonta. These similarities include well-defined, uncovered, and elongated olfactory bulbs, partly exposed midbrain, and prominent petrosal lobules. The two later parameters call for further remarks. See below

Abstract : « Primitive aspects of the endocranial morphology in *Megalagus* include ... exposure of the midbrain »,

also page 7, line 153-154

page 9, line 194

page 10, line 214 and 237-238

page 14, line 317-322

Midbrain exposure is not a primitive (archaic is I think a better word) character. following Starck, D. 1963. "Freiliegendes Tectum mesencephali" ein Kennzeichen des primitiven Säugetiergehirns? *Zoologischer Anzeiger* 171: 350-359. It would be worth checking this publication, is it outdated or not taking into account recent and current studies ?

See also Kass, J.H., Collins, C.E. 2001. Variability in the sizes of brain parts. *Behavioral and Brain Sciences* 24(2) : 288-290.

Please look at this note especially concerning page 9, line 197-198 : « The colliculi are variably visible in extant lagomorph endocasts » visible or exposed ? and page 11, line 245-246 « exposure of the colliculi is known to vary intraspecifically (e.g., in *Microsypops annectens* and *Ischyromys typus*) (30, 34). »

Page 5, line 92: Is « the jugal arches » the correct term ? or zygomatic arch ?

Page 6, line 119: the temporal foramina are not indicated on figure 3 (should be tf) but on figure 2

Page 6, line 133-134 : « Excluding the petrosal lobules, the cerebellum of *Megalagus* is slightly narrower than the cerebrum »,

also page 9, line 203

page 11, line 253

About the volume of the petrosal lobules, be careful that the cast of the cerebellum is probably a good proxy for cerebellum shape and volume but in the case of the petrosal lobules of the paraflocculus, the putative proxy is the subarcuate fossa cast and subarcuate fossa size is not a reliable proxy for paraflocculus size :

« The correlation between the size of the petrosal lobule of the paraflocculus and the subarcuate fossa in adults is weak » (Sanchez-Villagra, M.R. 2002. The cerebellar paraflocculus and the subarcuate fossa in *Monodelphis domestica* and other marsupial mammals - ontogeny and phylogeny of a brain-skull interaction. *Acta Theriologica* 47(1) : 1-14)

It's worth checking whether this bony structure is endochondral, in which case the lack of strict relationship between the fossa and the nervous structure inside it simply results from the fact that the bone doesn't result from the ossification of a mesenchymal envelope covering the nervous structure (as it is the case for instance for the bones constituting the braincase)

Of interest concerning page 11, lines 247-254 and especially 251-252 : « The vermis and cerebellar hemispheres are generally similarly developed in *Megalagus* as in *Ischyromyidae* and *Plesiadapiformes* », the expansion of the cerebellar hemispheres appears to be related to the expansion of specific regions of the cerebral cortex ;

see Smaers et al. 2018 <https://doi.org/10.7554/eLife.35696.001>

Referee: 2

Comments to the Author(s)

Overall Impressions

In this submission, López-Torres and colleagues contribute a description of the fossilized endocast of a remarkably complete early Oligocene lagomorph skull (*Megalagus*). They

qualitatively and quantitatively compare the endocast of *Megalagus* to other extant and extinct lagomorphs, rodents, and primates. This study is straight-forward and succinct, and the importance of the fossil is clear.

The strengths of the study lie in the incredible temporal and systematic context of this well-preserved fossil, as well as the remarkable comparative dataset the authors have amassed. Endocasts of extant and fossil mammals are not trivial to reconstruct from CT data, and the authors have done meticulous work building their digital dataset.

There are several book-keeping details that the authors should attend to before the manuscript can be accepted for publication, some of which may require major revisions. In this regard, I have been a bit more detail-oriented than usual due to the global pandemic. My apologies if this review comes off as too pedantic—it is my intention to help ensure the highest-quality end product to match the effort that the authors have clearly put in.

Book-Keeping and conceptual issues to address throughout:

1. Sample sizes are never directly reported, and key evolutionary conclusions are drawn from boxplots of skewed sample sizes (e.g. 21 extant rodents vs. 7 extant lagomorphs, judging by Table S4a). More even sample sizes between extant groups (for which obtaining additional specimens is more feasible) would ensure robusticity of the results.
2. Many of the key figures that are central to the results and discussion are only present in the supplemental information. For example, Figures S1, S2, and S4 contain visual information that is crucial to interpretation of the comparisons. Presumably it is the comparative evolutionary morphology that makes the description of this endocast suitable for a broad audience such as that of Proceedings B. By including only figures of the *Megalagus* endocast in the body of the manuscript, the authors have displaced evidence of their descriptive arguments as well as the bulk of their work. I strongly encourage the authors to identify a format and/or make revisions to the figures that would allow more of the total body of morphological evidence to be included in the main body of the text.
3. Why is *Megalagus* compared with both extant and extinct Glires, but only fossil Euprimates? Numerous evolutionary arguments are made about the directionality of brain region shifts in the context of primate evolution, but no extant primate endocasts are quantified.
4. Why are different taxonomic samples used for the boxplots, biplots, and PCA? E.g. extant rodents are absent from the biplot and PCA, but the reason this exclusion is never explained.

Specific Manuscript Sections:

Introduction:

Introduction is succinct and provides the necessary systematic and temporal context needed to demonstrate the import of the *Megalagus* endocast to interpretation of Euarchontoglires evolution. Figure 1 (phylogenetic scheme) is helpful and may be improved if revised as a phylogeny that annotates primitive and derived traits (see “Comparison with extant and fossil Lagomorpha” section below).

Description

A reference to a nomenclatural convention would enhance the reader’s ability to evaluate the appropriateness of structure identification in the description. This is especially critical in comparative neuroanatomy, where terms coined for human brain anatomy have been variably applied in zoological contexts. A standardized ontology would allow the reader to follow designations based on structural associations, homology, or both. The *Nomina Anatomica Veterinaria* and Swanson’s Brain Maps come to mind as having helpful ontologies, and the

authors have an extensive body of previous work and endocast evolution reviews they could refer to as well.

Throughout: Please check to ensure that figure references are placed after the described morphology rather than before, e.g. Line 96: "The endocast is elongate, with well-developed (Fig. 3), ovoid, and pedunculated olfactory bulbs (Fig. 3)..."

Line 108: From the figure provided (Fig. 3), I agree with the authors that there is no apparent Sylvian sulcus in *Megalagus*; however, I am not sure I confidently agree with their interpretation of the rostral extent of the rhinal fissure as a Sylvian fossa. Providing a reference definition for the Sylvian fissure/fossa based on e.g. structural associations would ensure agreement.

Lines 110–115: Clarification is needed regarding superior sagittal sinus/sulcus morphology: In the discussion, the authors point out that "The superior sagittal sinus is visible in the caudal half of the endocast of *Megalagus*, whereas in the rostral half there is a superior sagittal sulcus, but the sinus is not apparent..." However, "superior sagittal sinus" is labeled rostrally in Figure 3, near the olfactory bulbs, contradicting the text.

Comparisons with extant and fossil Lagomorpha

I understand that Proceedings B has significant page restrictions and financial penalties for going over a 6-page limit; however, the authors have an extremely helpful (and beautiful!) comparative endocast figure in their supplemental document. I would strongly suggest trying to find a way to include comparative figure S1 in the main body of the text.

The discussion of primitive characters displayed by *Megalagus* is a vital component to the manuscript--inclusion of the comparative figure S1 into the main body of text would greatly enhance the interpretability of the manuscript. In addition to including the anatomical figure, highlighting the traits discussed on a phylogeny would help readers interpret similarities between *Megalagus* and, for example, leporids vs. ochotonids.

Line 187: "The brain in all lagomorph taxa studied here is lissencephalic": should "brain" read "endocast" as the brain in *Megalagus*, a lagomorph taxon, is unavailable?

Discussion

Lines 290–292: "With respect to the olfactory bulbs, *Megalagus* is similar to plesiadapiforms and early fossil rodents in their size relative to the rest of the brain, with relatively larger bulbs than observed in living lagomorphs, or fossil or living euprimates (Fig. 4C)."

This is an interesting pattern, especially in light of recent literature documenting tradeoffs in olfactory morphology with cerebral hemisphere volume and/or thermoregulatory structures in mammals and birds; however, I am concerned that the observed pattern may be an artifact of sample size. In figure 4C, the value for *Megalagus* olfactory bulb volume ratio only falls within the box plot ranges for fossil and extant rodents. All of the box plots that represent groupings within Glires appear to be relatively conserved, and the patterns of evolutionary brain region shifts the authors are discussing may be driven by an imbalance in sample size rather than a real evolutionary pattern. From Table S4a, there are 300% more extant rodents than extant lagomorphs (note that I was unable to find any direct reporting of sample sizes in the main text or supporting information and counted from the table). The low amount of variance in lagomorphs and ochotonids compared to the rodents may be driven by the low sample size (N= 7 & 3 respectively). The authors should include additional lagomorph samples to ensure that their conclusions are robust to sample size equalization. Finally, comparisons with plesiadapiforms may be over-stated if this within-Glires pattern fails to hold up to additional specimens.

Lines 304–306: “In *Megalagus* the anterior extremities of its olfactory bulbs reach the area over M1; thus, they are more posteriorly located in the cranium compared to modern lagomorphs (up to P3–P4; Fig. S2).” Parenthetical needs reference to Fig. 2i so comparison can be visualized.

Tables, Figures, & Captions:

Table 1: The measurement table is extremely helpful, and the tables in the supplemental provide a wealth of information. However, the measurements are not illustrated on a specimen, nor are anatomical landmark criteria for the measurements provided. Minimally, such a figure is necessary in the supplemental, and ideally the authors might also include landmark criteria (either written into the text or referred to a source).

Comparing Table 1 with Table S3: Why was olfactory bulb height measured in *Megalagus* but omitted from the PCA measurements?

Figure 3:

Please list structure abbreviations in alphabetical order and ensure that all structures are divided by consistent punctuation. Color coding of labels should be explained.

Caption is missing abbreviation for: “cer,” “CN XII”, “lat-si,” “midb,” “olf-bu,” “Sy-fo”

The structures “lat-si” in dorsal view and “lat-su” in lateral view appear to be the same structure. A single naming convention should be adhered to in the figure and caption.

Figure 4:

Both fossil and extant rodents are included in the box plots, but extant rodents do not appear to be included in the bivariate plot. These specimens should be included or a reason should be provided for why they are not. The sample size for the number of specimens in each box plot is not directly reported in the main text or supplemental.

Figure 5:

Why are extant rodents omitted from the PCA? The color-coding for groups in the PCA should be made consistent with the color coding in the box plots and biplot.

Supporting Information:

Materials & Methods:

Scanning and rendering methods for *Megalagus* are adequate. Scan parameters should also be provided for the comparative specimens, if even in the form of providing the morphosource links.

Fig. S1: Naming conventions are not standard throughout the figure caption (e.g. Leporidae is pointed but, but Ochotonidae is not; color-coding is not described). Anatomical labels are needed to satisfactorily compare other lagomorph endocasts with the *Megalagus* endocast.

Fig. S6: Anatomical labels are needed to satisfactorily compare Euarchontoglires endocasts with the *Megalagus* endocast.

Fig. S7: From the manuscript text, it is implied that the numerical features in the loading plots correspond with the order that the features are listed in Table S3 (i.e. loading factor 1 is endocast TL). Please include confirmation of this in the Fig S7 caption.

Table S4a: Caption needs correction: “Data used for the box plot analyses in Figure 4B, 4C, and 4C.” In the Group column of the table, specify that Leporidae and Ochotonidae specimens are extant and Euprimates are extinct to maintain naming conventions of the main text.

Referee: 3

Comments to the Author(s)

The paper makes a significant contribution to the field of paleobiology by interpreting the neuroanatomy of a fossil that fills a critical gap in our knowledge of gliran and lagomorphan brain evolution. The authors do a good job of describing the condition that they observe in the fossil taxon. One area that could be improved is in contextualizing the fossil more by summarizing major changes in neuroanatomy within Lagomorpha in an evolutionary and ecological context. While there is certainly a good discussion of similarities between taxa, what is missing is a broader context for those similarities. Similarly, while there are many functional arguments one could make from the neuroanatomy of this taxon, the authors seem hesitant to make them, which seems like an intentional oversight. If that is the intention, a reason should be given for that omission. Overall, I think the paper is important and worthy of publication. I think your figures are engaging, easy to follow, and informative. I look forward to seeing it published.

Line 72 - The authors state that the "the evolutionary history of the lagomorph brain is almost entirely unknown" making this study quite critical for addressing that knowledge gap however then raises the question of why they didn't discuss evolutionary patterns along lagomorpha in the following text.

85 - For those not familiar with Glires and Euarchonta, it is not apparent why we should expect the basal condition to be different from the recent condition or indeed if the clades should be expected to differ at all. Why is it important to consider the primitive condition here? Are there major changes within Glires that this new specimen may inform the timing of?

102 - You may consider including a brief definition of lissencephalic

148 - I believe the authors should include a brief summary of trends among the comparative lagomorph sample, clearly defining what the primitive conditions in the clade are. It is difficult to follow the evolutionary context of the fossil specimen without general trends among lagomorphs clearly laid out.

Figure 2 is really quite lovely. You might consider rotating the lateral view of the endocast in the skull (i) slightly counter-clockwise so that it's at the same orientation as (c).

Within the Material and Methods, you should consider including a project number and DOI for surface files used in this project.

Fig. S3A, the Y-axis title is misspelled

Author's Response to Decision Letter for (RSPB-2020-0665.R0)

See Appendix A.

Decision letter (RSPB-2020-0665.R1)

29-May-2020

Dear Dr Fostowicz-Frelik

I am pleased to inform you that your manuscript entitled "Cranial endocast of the stem lagomorph *Megalagus* and brain structure of basal Euarchontoglires" has been accepted for publication in Proceedings B. Congratulations!!

Open Access

Your article has been estimated as being 9 pages long. Our Production Office will be able to confirm the exact length at proof stage.

Paper charges

Sincerely,

Dr John Hutchinson
Editor, Proceedings B
<mailto:proceedingsb@royalsociety.org>

Appendix A

ASSOCIATE EDITOR

We are really grateful for your comments and very thorough reviews provided by the referees. Please find our detailed answers below.

Corrections and additions to main text and figures

We took seriously Reviewer 2's recommendation and extended Fig. 3 by adding representatives of extant leporids and ochotonids to facilitate immediate morphological comparisons. Because we had to reduce the picture of *Megalagus* endocast, we decided to include original Figure 3 into our Supplementary material (it is now Fig. S2).

Furthermore, we incorporated Fig. S4 into Fig. 4 (as Fig. 4F).

We also considerably improved Fig.1 along with Reviewers suggestion.

We checked also the References section and fixed some minor problems (e.g. ref. 3 was missing).

Finally, we deposited the surface rendering of *Megalagus* endocast in the Dryad Digital Repository.

The dataset identifier is doi:10.5061/dryad.0vt4b8gwg. We cite this dataset in the main text and list it in the references.

Corrections and additions to Supplementary material

We submit this material as a single pdf file.

We added Table S1 with information on lagomorph micro-CT scans used in our paper; thus, the rest of tables were renumbered accordingly.

We added two new figures: Fig. S1, a guide to the linear measurements of endocast, and Fig. S2, with digital endocast of *Megalagus turgidus*. The latter is an enhanced and corrected (minor labelling issues) version of former Fig. 3 from the main text.

We reorganized Fig. S1 of our original submission (which is now Fig. S3) to enable easier comparisons between the lagomorph taxa.

We moved Fig. S4 to the main text as advised by Reviewer 2. It is now a part of Fig. 4 (Fig. 4F).

REVIEWER 1

Primitive aspects of the endocranial morphology in *Megalagus* include ... exposure of the midbrain », also page 7, line 153-154, page 9, line 194, page 10, line 214 and 237-238, page 14, line 317-322 . Midbrain exposure is not a primitive (archaic is I think a better word) character. Following Starck, D. 1963. "Freiliegendes Tectum mesencephali" ein Kennzeichen des primitiven Säugetiergehirns? Zoologischer Anzeiger 171: 350–359. It would be worth checking this publication, is it outdated or not taking into account recent and current studies ?

We agree that 'archaic' is a more appropriate term for the exposed midbrain condition than 'primitive'. It is most probably a plesiomorphic (ancestral) character for Eutheria. The survey on

midbrain exposure in different groups of extant placentals shows a highly mosaic picture (e.g., among Chiroptera).

Nevertheless, in the context of our paper we are centered on the extent of midbrain exposure in Euarchontoglires (both stem and crown groups). From such point of view, an exposed or partly exposed midbrain is observed in most of the Euarchontoglires basal taxa (*Megalagus* included), whereas a covered midbrain is generally typical of crown taxa. Therefore, considering the extinct and extant Euarchontoglires ‘the exposed or partly exposed midbrain’ certainly may be recognized also as a primitive feature.

To sum up, we are now more precise in our wording.

See also Kass, J.H., Collins, C.E. 2001. Variability in the sizes of brain parts. Behavioral and Brain Sciences 24(2): 288-290. Please look at this note especially concerning page 9, line 197-198: The colliculi are variably visible in extant lagomorph endocasts visible or exposed ? and page 11, line 245-246 exposure of the colliculi is known to vary intraspecifically (e.g., in *Microsypops annectens* and *Ischyromys typus*) (30, 34).

The colliculi are the structural parts of the midbrain, they are delicately marked at the tectum, and visible only when the midbrain is well-exposed. In case of the endocasts specifically, we can talk only about the visibility of these structures as it is the only information we can actually get from the endocast. The visibility of the colliculi can be strongly dependent on many factors, including the diagenetic conditions or the overall state of the skull preservation.

We also know from, our experience, that the presence or absence of the colliculi, even in extant mammalian endocasts, can be intra-specifically variable, e.g., we know that pikas (*Ochotona*) and rabbits (*Oryctolagus cuniculus*) have the colliculi developed, but these structures may not be visible at the endocast nonetheless.

We carefully studied the papers suggested by Reviewer 1; nevertheless, they are not directly relevant our paper. We agree that in Euarchontoglires the size of the colliculi is rather variable, although their presence and their larger size is generally associated with the endocasts of basal taxa. However, Silcox et al. (2009) suggested that enlarged colliculi in extant Dermoptera may be a secondary character. In case of *Megalagus*, the colliculi are indiscernible at the endocast, thus we cannot interpret neither their size nor morphology.

Page 5, line 92: the ‘jugal arches’

We corrected this term to ‘zygomatic arches’.

Page 6, line 119: the temporal foramina are not indicated on figure 3

We marked them on Figures 2 and 3. We also added Fig. S2, which is a larger image of the *Megalagus* endocast and the temporal foramina are also marked there.

Page 6, line 133-134, also page 9, line 203, page 11, line 253, concerning the size and content of the petrosal lobules and development of the subarcuate fossa

In regard to the connection between the paraflocculi and subarcuate fossa for Euarchontoglires (and Primates in particular), we can briefly answer: “Results show that, in mammals, the size and morphology of the petrosal lobule is directly related to that of the subarcuate fossa.” [see Gannon, P. J., Eden, A. R., & Laitman, J. T. (1988). The subarcuate fossa and cerebellum of extant primates: Comparative study of a skull-brain interface. *American Journal of Physical Anthropology*, 77(2), 143–164.].

Reviewer 1 quotes Sanchez-Villagra (2002) paper on marsupials. We prefer to follow the paper by Gannon et al. (1988) which concerns nine orders of mammals, including rodents, lagomorphs, and primates, as well as marsupials, although it is focused on primates. Sanchez-Villagra (2002) suggests that Ganon et al. (1988) did not perform adequate statistical testing for the relationships between the subarcuate fossa and the petrosal lobules and the lack of relationship between the two structures in marsupials means that this relationship should be reexamined in future, with which we agree. As there is very little known on the matter of the subarcuate fossa/petrosal lobule relations in Euarchontoglires, we prefer to follow Gannon et al. (1988) in this respect, since our study is related directly to placentals, until more exhaustive study is published.

Concerning the Reviewer 1’s question on the development and endochondral origin of the subarcuate fossa: There is very little known on the subject. What is actually known, it is that subarcuate fossa is not formed via ossification of tissue surrounding the paraflocculi. Instead, the formation of the subarcuate fossa is connected to the growth and development of the semi-circular canals, and the petrosal lobules secondarily occupy the fossa (McClure and Daron (1971). The relationship of the developing inner ear, subarcuate fossa and paraflocculus in the rat. *American Journal of Anatomy*, 130(2), 235–249).

This, however, does not mean that the size of the subarcuate fossa is only a product of variation in the size of the semicircular canals as the size of the fossa appears to contribute significantly the variation in the size of the anterior canal and width of the posterior canal (Jeffery, N., Ryan, T. M., & Spoor, F. (2008). The primate subarcuate fossa and its relationship to the semicircular canals part II: Adult interspecific variation. *Journal of Human Evolution*, 55(2), 326–339).

Page 11, lines 247-254, and especially 251–252

We agree that it would be very interesting to dwell more on the subject of particular adaptations of *Megalagus*. The paper by Smaers et al. (2018) concerns directly the development of cognition related to the lateral cerebellar expansion. This certainly did not happen in early Glires or stem primates, as they show similar and rather limited level of the cerebellar development. Nevertheless, it is interesting question for further studies.

REVIEWER 2

Book-Keeping and conceptual issues to address throughout

1. Sample sizes are never directly reported, and key evolutionary conclusions are drawn from boxplots of skewed sample sizes (e.g. 21 extant rodents vs. 7 extant lagomorphs, judging by Table S4a).

We understand the necessity of reporting sample sizes, so we are now reporting them in the figure captions, whenever boxplots are used (i.e., Figs. 4 and S9).

By “skewed” sample sizes in this context Reviewer 2 means that there are disproportions in sample sizes (rodents seem overrepresented). However, we would like to note that with regard to the lagomorph vs. rodent sample issue, the other argument could actually be made. As a total of the overall diversity, our sampling of Lagomorpha (7 out of ca. 95 extant species, completeness – 7.4%) is actually more complete than our sampling of Rodentia (21 out of ca. 2400 extant species, completeness – 0.88%). Thus, adding more lagomorphs would not be a very good idea. Anyway, our lagomorph sample captures representatives of the main lineages and ecologies (leporids vs. ochotonids; among leporids, *Romerolagus diazi* represents an earlier radiation).

Furthermore, because of the disparities in sample sizes, we specifically used box plots to estimate the statistical differences. In this case, the median (50th percentile of variable) and interquartile range (the central 50% of the data) are appropriate measures of central tendency and variability, respectively, and thus, our evolutionary conclusions drawn from them will hold also for less disparate samples.

Finally, “more even” sample sizes of extant Rodentia vs. Lagomorpha are hard to be achieved also at a higher (generic) level. Six genera of Lagomorpha in our sample constitute 50% of all living genera, and for a proportional ratio in Rodentia we would have to sample ca. 230 genera of extant Rodentia, which is hardly feasible.

2. Many of the key figures that are central to the results and discussion are only present in the supplemental information. For example, Figures S1, S2, and S4 contain visual information that is crucial to interpretation of the comparisons [...]. I strongly encourage the authors to identify a format and/or make revisions to the figures that would allow more of the total body of morphological evidence to be included in the main body of the text.

In principle, we agree. However, we have to bear in mind also the article length allowed by the journal (and this is a critical factor here). That said, we modified Fig. 2 in such a way that it now includes *Ochotona* and *Lepus* (the representatives of the only two extant families of Lagomorpha), important for comparisons.

We have merged Fig. S4 with Fig.4, so all information crucial for comparisons is now in the main text.

3. Why is *Megalagus* compared with both extant and extinct Glires, but only fossil Euprimates? Numerous evolutionary arguments are made about the directionality of brain region shifts in the context of primate evolution, but no extant primate endocasts are quantified.

The point here is that our paper is about early Euarchontoglires. Thus, we do not add data from extant primates. We know there is a significant shift in the form of the brain from plesiadapiforms to Euprimates (see Harrington et al. 2016), and including data from the diversity (in brain and body size) of living primates in our paper is going to muddy the picture.

We believe that fossil euprimates in our sample help frame our study better around the primitive nodes of Euarchontoglires. Our comparative sample comprises groups that are either very closely related to stem lagomorphs (i.e., modern lagomorphs, and fossil and modern rodents; as members of Glires) or that come from the Paleogene but are (or have been argued to be) still Euarchontoglires (i.e., plesiadapiforms, fossil euprimates, and apatemyids). So, fundamentally, the further away we get from a stem lagomorph, the less ideal a group is for comparison.

4. Why are different taxonomic samples used for the boxplots, biplots, and PCA? E.g. extant rodents are absent from the biplot and PCA, but the reason this exclusion is never explained.

In most cases, this is simply because the information available (e.g., from the literature) varies depending on the endocast metric being studied. For example, the neocortical ratio has never been calculated for apatemyids, or the petrosal lobule volume ratio for fossil euprimates, plesiadapiforms, and apatemyids.

In the specific case of Figs 4E and 5, extant rodents were not included because they occupied the entire morphospace of the other groups combined. We originally ran the analysis with extant rodents included and, in the context of the biplot and PCA, they 1) did not provide any information that would help discriminate between groups, and 2) they made the plot harder to read because it became too cluttered.

See also our general point on modern Rodentia (Comment 1).

Specific Manuscript Sections

Introduction:

5. Introduction is succinct and provides the necessary systematic and temporal context needed to demonstrate the import of the *Megalagus* endocast to interpretation of Euarchontoglires evolution. Figure 1 (phylogenetic scheme) is helpful and may be improved if revised as a phylogeny that annotates primitive and derived traits (see “Comparison with extant and fossil Lagomorpha” section below).

We agree that Reviewer 2’s suggestion would improve Fig. 1, so we have now included annotations for primitive and derived traits on the phylogenetic scheme.

Description:

6. A reference to a nomenclatural convention would enhance the reader's ability to evaluate the appropriateness of structure identification in the description. This is especially critical in comparative neuroanatomy, where terms coined for human brain anatomy have been variably applied in zoological contexts. A standardized ontology would allow the reader to follow designations based on structural associations, homology, or both. The Nomina Anatomica Veterinaria and Swanson's Brain Maps come to mind as having helpful ontologies, and the authors have an extensive body of previous work and endocast evolution reviews they could refer to as well.

We have now indicated that we are following Silcox et al. (2009) in the Materials and Methods section. However, we are using the term "petrosal lobule" instead of "paraflocculus".

Concerning 'a standardized ontology', we do not think that cross-referencing between the NAVI (or Professor Swanson's work) and our paper is really necessary.

7. Throughout: Please check to ensure that figure references are placed after the described morphology rather than before, e.g. Line 96: "The endocast is elongate, with well-developed (Fig. 3), ovoid, and pedunculated olfactory bulbs (Fig. 3)..."

We corrected this issue.

8. Line 108: From the figure provided (Fig. 3), I agree with the authors that there is no apparent Sylvian sulcus in Megalagus; however, I am not sure I confidently agree with their interpretation of the rostral extent of the rhinal fissure as a Sylvian fossa. Providing a reference definition for the Sylvian fissure/fossa based on e.g. structural associations would ensure agreement.

We agree with Reviewer 2. What we had indicated as a Sylvian fossa in Fig 3 may not be actually a Sylvian fossa. We have removed that label from Figure 3 and the pertinent text of the manuscript.

9. Lines 110–115: Clarification is needed regarding superior sagittal sinus/sulcus morphology: In the discussion, the authors point out that "The superior sagittal sinus is visible in the caudal half of the endocast of Megalagus, whereas in the rostral half there is a superior sagittal sulcus, but the sinus is not apparent..." However, "superior sagittal sinus" is labeled rostrally in Figure 3, near the olfactory bulbs, contradicting the text.

We agree with Reviewer 2. The area labelled as sus-si (superior sagittal sinus) in Fig. 3 is indeed the superior sagittal sulcus; it has been corrected. We have also added another label indicating the superior sagittal sinus in the caudal half for more clarity.

Comparisons with extant and fossil Lagomorpha:

10. I understand that Proceedings B has significant page restrictions and financial penalties for going over a 6-page limit; however, the authors have an extremely helpful (and beautiful!) comparative endocast figure in their supplemental document. I would strongly suggest trying to find a way to include comparative figure S1 in the main body of the text.

We have addressed this issue in the response to Comment 2.

11. The discussion of primitive characters displayed by Megalagus is a vital component to the manuscript—inclusion of the comparative figure S1 into the main body of text would greatly enhance the interpretability of the manuscript. In addition to including the anatomical figure, highlighting the traits discussed on a phylogeny would help readers interpret similarities between Megalagus and, for example, leporids vs. ochotonids.

We agree. Please see our response to Comments 2 and 5.

12. Line 187: “The brain in all lagomorph taxa studied here is lissencephalic”: should “brain” read “endocast” as the brain in Megalagus, a lagomorph taxon, is unavailable?

We corrected the phrase to avoid any misunderstanding.

Discussion:

13. Lines 290–292: “With respect to the olfactory bulbs, Megalagus is similar to plesiadapiforms and early fossil rodents in their size relative to the rest of the brain, with relatively larger bulbs than observed in living lagomorphs, or fossil or living euprimates (Fig. 4C).” This is an interesting pattern, especially in light of recent literature documenting tradeoffs in olfactory morphology with cerebral hemisphere volume and/or thermoregulatory structures in mammals and birds; however, I am concerned that the observed pattern may be an artifact of sample size. In figure 4C, the value for Megalagus olfactory bulb volume ratio only falls within the box plot ranges for fossil and extant rodents.

We checked and corrected this. We are talking about “olfactory bulb length ratio”, not “volume ratio”, and our observation still holds (see Table S4).

All of the box plots that represent groupings within Glires appear to be relatively conserved, and the patterns of evolutionary brain region shifts the authors are discussing may be driven by an imbalance in sample size rather than a real evolutionary pattern. From Table S4a, there are 300% more extant rodents than extant lagomorphs (note that I was unable to find any direct reporting of sample sizes in the main text or supporting information and counted from the table). The low amount of variance in lagomorphs and ochotonids compared to the rodents may be driven by the low sample size (N= 7 & 3 respectively). The authors should include additional lagomorph samples to ensure that their conclusions are robust to sample size equalization.

Finally, comparisons with pleasiadaptiforms may be over-stated if this within-Glires pattern fails to hold up to additional specimens.

Now, we do not exactly see eye to eye with Reviewer 2. In fact, for the reasons provided earlier (the response to Comment 1), our lagomorph sample is fairly representative of the entire order. Indeed, having included members of radically different lagomorph lineages in our sample gives us confidence to suggest that we are seeing a representation of the real diversity in our plots.

14. Lines 304–306: “In *Megalagus* the anterior extremities of its olfactory bulbs reach the area over M1; thus, they are more posteriorly located in the cranium compared to modern lagomorphs (up to P3–P4; Fig. S2).” Parenthetical needs reference to Fig. 2i so comparison can be visualized.

We added this.

Tables, Figures, & Captions:

15. Table 1: The measurement table is extremely helpful, and the tables in the supplemental provide a wealth of information. However, the measurements are not illustrated on a specimen, nor are anatomical landmark criteria for the measurements provided. Minimally, such a figure is necessary in the supplemental, and ideally the authors might also include landmark criteria (either written into the text or referred to a source).

We agree. We have prepared and included a relevant figure (Fig. S1) in the Supplementary Material.

16. Comparing Table 1 with Table S3: Why was olfactory bulb height measured in *Megalagus* but omitted from the PCA measurements?

The reason is not enough data on this character from other taxa.

Figure 3:

17. Please list structure abbreviations in alphabetical order and ensure that all structures are divided by consistent punctuation. Color coding of labels should be explained.

The abbreviations are now alphabetized, and the color coding is now explained.

18. Caption is missing abbreviation for: “cer,” “CN XII,” “lat-si,” “midb,” “olf-bu,” “Sy-fo”.

These abbreviations are now explained or, in case of ‘Sy-fo’, removed.

19. The structures “lat-si” in dorsal view and “lat-su” in lateral view appear to be the same structure. A single naming convention should be adhered to in the figure and caption.

That was a typo. Both should say “lat-su”, lateral sulcus. It has now been corrected.

Figure 4:

20. Both fossil and extant rodents are included in the box plots, but extant rodents do not appear to be included in the bivariate plot. These specimens should be included or a reason should be provided for why they are not. The sample size for the number of specimens in each box plot is not directly reported in the main text or supplemental.

See the responses to Comments 1 and 4 by Reviewer 2. Alas, this paper is not centered on Rodentia.

Figure 5:

21. Why are extant rodents omitted from the PCA? The color-coding for groups in the PCA should be made consistent with the color coding in the boxplots and biplot.

See the response to Comment 4.

Supporting Information:

Materials & Methods:

22. Scanning and rendering methods for *Megalagus* are adequate. Scan parameters should also be provided for the comparative specimens, if even in the form of providing the morphosource links.

We agree. We have provided these scan parameters in a new table (Table S1).

23. Fig. S1: Naming conventions are not standard throughout the figure caption (e.g. *Leporidae* is pointed but, but *Ochotonidae* is not; color-coding is not described). Anatomical labels are needed to satisfactorily compare other lagomorph endocasts with the *Megalagus* endocast.

We are now consistent with the naming conventions, we now explain the color-coding, and we have added labels to the figure.

24. Fig. S6: Anatomical labels are needed to satisfactorily compare *Euarchontoglires* endocasts with the *Megalagus* endocast.

We beg to differ. All necessary anatomical labels are in Fig. S3. Including them in Fig. S7 (formerly Fig. S6) would unnecessarily cram the illustration, which is not after all a tutorial on brain morphology.

REVIEWER 3

Comments to the Author(s):

[...] **One area that could be improved is in contextualizing the fossil more by summarizing major changes in neuroanatomy within Lagomorpha in an evolutionary and ecological context.**

We agree with Reviewer 3 on that. We have provided improved Fig. 1, where we included annotations for primitive and derived traits on the phylogenetic scheme, thus summarizing major neurobiological changes. However, we are a bit reluctant to make any strong statements on ecological implications, because this is the first digital endocast of fossil lagomorphs ever. We certainly will be gathering more data on Lagomorpha (both extinct and living representatives of the order) to fully explore these issues later.

Line 72: [...] then raises the question why they didn't discuss evolutionary patterns along Lagomorpha in the following text.

First, this study concerns the evolutionary patterns within basal Euarchontoglires, so the proposed topic is 1) a bit specialized for this paper, and 2) we do not have information on other fossil lagomorphs as yet. To explore fully evolutionary patterns across Lagomorpha, we would need endocast data from other fossil genera (at least near the leporid–ochotonid split), although obviously we cannot expect as good coverage as achieved by Fostowicz-Frelik and Meng (2013) for the premolar foramen. In our opinion, to discuss general evolutionary patterns in Lagomorpha at present would be premature. As we said earlier, the size of our sample will certainly be increased in future studies as more lagomorph crania are available.

Line 85 - For those not familiar with Glires and Euarchonta it is not apparent why we should expect the basal condition to be different from the recent condition or indeed if clades should be expected to differ at all [...].

As noted e.g. by Jerison (1973) there is the tendency toward encephalization and neocorticalization increase during evolution in all mammalian groups, especially visible in the Primates. Other Euarchontoglires groups are no exception here. Having data on the early representatives (Paleogene fossil record) of stem primates and basal rodents we know that they differed in neuroanatomy from the living members of their respective orders. Because lagomorphs are known for their conservative morphology, we were curious if their early member *Megalagus* would follow the same pattern and how its basic brain structure (or, more precisely, the cranial endocast) was different that of extant rabbits, hares, and pikas. In other words, what the primitive lagomorph character-set is.

102 - You may consider including a brief explanation of lissencephalic

We added 'smooth' in the text (we are talking about endocasts here).

148 - I believe the authors should include a brief summary of trends among the comparative lagomorph sample defining what the primitive conditions in the clade are.

We specified clearly the primitive character-set for Lagomorpha (see Fig. 1).

Figure 2

We prefer not to rotate Fig. 2. Our intention is that the lateral view of the endocast be parallel to the skull orientation in the life restoration of *Megalagus*.

Within the Material and Methods, you should consider including a project number and DOI for surface files used in this project.

We agree. We deposited the surface renderings of the *Megalagus* endocast in the Dryad Digital Repository. The dataset identifier is doi:10.5061/dryad.0vt4b8gwg.

Fig. S3A, the Y-axis title is misspelled

We corrected this (now in Figure S5).